# Synthesis and Biological Activity of a VHL-Based PROTAC Specific for p38α

**DOI:** 10.3390/cancers15030611

**Published:** 2023-01-18

**Authors:** Mónica Cubillos-Rojas, Guillem Loren, Yusuf Z. Hakim, Xavier Verdaguer, Antoni Riera, Angel R. Nebreda

**Affiliations:** 1Institute for Research in Biomedicine (IRB Barcelona), The Barcelona Institute of Science and Technology, Baldiri Reixac 10, 08028 Barcelona, Spain; 2Department Química Inorgànica i Orgànica, Universitat de Barcelona, Martí i Franquès 1, 08028 Barcelona, Spain; 3ICREA, Pg. Lluís Companys 23, 08010 Barcelona, Spain

**Keywords:** p38 MAPK, PROTAC, protein degradation, VHL

## Abstract

**Simple Summary:**

The compounds named PROTACs are formed by two fragments, which bring together a particular protein with a ubiquitinating enzyme. This allows ubiquitination and degradation of the targeted protein. Ubiquitination-mediated protein degradation is an important regulatory process to control the expression levels of proteins and maintain the homeostatic conditions in cells. Thus, by re-directing a mechanism that is normally used for cell regulation, PROTACs allow to remove specific proteins associated to particular diseases or pathological conditions. In this paper, we report a type of PROTAC that targets for degradation the protein named p38α, whose functions have been linked to cancer and other diseases. We show that these PROTACs effectively reduce p38α protein expression not only in several cancer cell lines but also in tumors generated in mice. These compounds may provide an attractive strategy to evaluate potential therapeutic applications for targeting p38α in the clinical context.

**Abstract:**

We report a series of small molecule proteolysis-targeting chimeras (PROTACs) that target the protein kinase p38α for degradation. These PROTACs are based on a ligand of the VHL E3 ubiquitin ligase, which is linked to an ATP competitive inhibitor of p38α. We provide evidence that these compounds can induce the specific degradation of p38α, but not p38β and other related kinases, at nanomolar concentrations in several mammalian cell lines. We also show that the p38α-specific PROTACs are soluble in aqueous solutions and therefore suitable for their administration to mice. Systemic administration of the PROTACs induces p38α degradation only in the liver, probably due to the PROTAC becoming inactivated in that organ, but upon local administration the PROTACs induce p38α degradation in mammary tumors. Our compounds provide an alternative to traditional chemical inhibitors for targeting p38α signaling in cultured cells and in vivo.

## 1. Introduction

Cells use signaling pathways to interpret the cues from their environment and respond accordingly. p38 mitogen-activated protein kinases (MAPKs) are at the core of several pathways involved in cellular responses to a broad variety of stimuli, including environmental stresses and inflammatory cytokines [1,2,3,4].

The mammalian p38 MAPK family comprises four members: p38α, p38β, p38γ and p38δ. The most abundant family member is p38α, which is ubiquitously expressed in almost all cell types, whereas p38β is usually expressed at lower levels, except in some brain regions. Conversely, p38γ is highly expressed in skeletal muscle, and p38δ is detected in the pancreas, kidney and small intestine [2,4,5]. Mice deficient for p38α have embryonic lethality due to placental defects [6,7], but single knockouts for p38β, p38γ or p38δ do not result in abnormalities during mouse development [8,9]; however, the double KO for p38α and p38β show additional phenotypes, suggesting that these two p38 MAPKs may have compensatory functions [4,10].

Upon stimulus-induced activation, p38α can phosphorylate a wide range of substrates, including protein kinases, transcription factors and membrane receptors, participating in the regulation of many cellular processes such as proliferation, differentiation and survival. Depending on the stimuli, the intensity and duration of the signal and the cell type, different outcomes can be triggered.

Dysregulated p38α activity has been associated with several human pathologies that include cardiovascular, neurodegenerative or inflammatory diseases as well as cancer, making it an attractive therapeutic target [2,4,11,12]. For this reason, many inhibitors have been developed and advanced through clinical trials, although none has progressed beyond phase III, mostly due to undesirable side effects or lack of efficacy in long term treatments [12,13,14].

Proteolysis targeting chimeras (PROTACs) are heterobifunctional compounds that induce the degradation of a protein of interest by promoting the interaction between the target protein and an endogenous E3 ligase [15]. This interaction induces the ubiquitination of the target and its subsequent degradation by the proteasome. A PROTAC is composed of two warheads, one that binds the target and the other for the E3 ligase, separated by a chemical linker. In contrast with other small-molecule drugs, PROTACs act via a catalytic mechanism, which might overcome some of the drawbacks associated with traditional inhibitions, such as the reduction of potency of the inhibitor due to mutations on the active site observed in cancer [15,16,17].

We previously reported a series of p38α PROTACs based on thalidomide to recruit the Cereblon (CRBN) E3 ligase and PH-797804, an ATP competitive inhibitor of p38α and p38β as warheads. These CRBN-based PROTACs induce a consistent degradation of both p38α and p38β proteins in several mammalian cell lines at nanomolar concentrations [18]. Although very potent and specific, these compounds were poorly soluble, which precludes their use in animal models. Here we report the design, synthesis and characterization of a new family of PROTACs based on a ligand for the E3 ligase VHL, which efficiently target p38α for degradation in mammalian cell lines. These VHL-based PROTACs show a dramatically increased solubility that allows their administration to mice.

## 2. Materials and Methods

### 2.1. Chemistry

Compounds **NR-11a–c** and **NR-11c*** were synthetized by coupling alkyne **1** with azides **2a–c** or **2c*** by a copper catalyzed azide-alkyne cycloaddition (CuAAC) (Figure 1). Alkyne **1** was prepared according to our previous p38 PROTAC synthesis [18]. Azides **2a–c** and **2c*** were prepared according to our previous publication [19].

### 2.2. General Method CuCAAC

A flask charged with the alkyne (1 eq.), the corresponding azide (1 eq.), CuSO_4_ (0.1 eq.), sodium ascorbate (0.2 eq.), H_2_O/*tert*-butanol 2:1 (0.026 M) and acetic acid (2 eq.) was sonicated 5 min and then stirred overnight at 30 °C. Brine and DCM were added to the reaction mixture, and the aqueous layer was extracted with DCM, dried over MgSO_4_ and concentrated under reduced pressure.

#### 2.2.1. (2*S*,4*R*)-1-((*S*)-2-(4-(4-(2-(3-(3-bromo-4-((2,4-difluorobenzyl)oxy)-6-methyl-2-oxopyridin-1(2H)-yl)-4-methylbenzamido)ethyl)-1H-1,2,3-triazol-1-yl)butanamido)-3,3-dimethylbutanoyl)-4-hydroxy-*N*-((*S*)-1-(4-(4-methylthiazol-5-yl)phenyl)ethyl)pyrrolidine-2-carboxamide (**NR-11a**)

Compound **NR-11a** was prepared following the general method for CuCAAC, starting from 3-(3-bromo-4-((2,4-difluorobenzyl)oxy)-6-methyl-2-oxopyridin-1(2H)-yl)-4-methyl-N-(pent-4-yn-1-yl)benzamide (**1**, 40 mg, 0.08 mmol) and (2*S*,4*R*)-1-((*S*)-2-(4-azidobutanamido)-3,3-dimethylbutanoyl)-4-hydroxy-*N*-((*S*)-1-(4-(4-methylthiazol-5-yl)phenyl)ethyl)pyrrolidine-2-carboxamide (**2a**, 43 mg, 0.08 mmol). The crude was purified by flash column chromatography, eluting with 7% MeOH/DCM. Compound **NR-11a** was obtained as a white solid (45 mg, 0.04 mmol, 50% yield).

^1^H NMR (400 MHz, CD_3_OD) δ 8.86 (s, 1H), 8.60–8.47 (m, 1H), 7.88 (dd, *J* = 12.0, 8.4 Hz, 2H), 7.77 (s, 1H), 7.69–7.57 (m, 2H), 7.53–7.43 (m, 1H), 7.40 (d, *J* = 1.3 Hz, 4H), 7.10–6.98 (m, 2H), 6.65 (s, 1H), 5.34 (s, 2H), 4.98 (dddq, *J* = 9.5, 7.1, 4.8, 2.4, 2.0 Hz, 1H), 4.58 (td, *J* = 8.4, 4.7 Hz, 2H), 4.41 (dt, *J* = 4.2, 2.1 Hz, 1H), 4.37 (t, *J* = 6.8 Hz, 2H), 3.89 (d, *J* = 11.0 Hz, 1H), 3.73 (dd, *J* = 11.0, 3.9 Hz, 1H), 3.39 (d, *J* = 6.9 Hz, 2H), 2.75 (q, *J* = 12.6, 10.0 Hz, 2H), 2.46 (s, 3H), 2.38–2.08 (m, 5H), 2.07 (d, *J* = 3.4 Hz, 3H), 2.01–1.84 (m, 6H), 1.48 (dd, *J* = 7.0, 2.4 Hz, 3H), 1.10–0.95 (m, 9H) (Appendix A).

^13^C NMR (101 MHz, CD_3_OD) δ 172.9, 171.8, 170.9, 167.1, 164.5, 163.4 (d, *J* = 242.2 Hz), 161.0, 160.8 (ddd, *J* = 243.2, 121.2, 12.3 Hz), 151.5, 147.2, 144.2, 139.2, 137.7, 133.8, 131.3, 131.2, 130.1, 129.1, 128.0, 126.6, 126.2, 126.0, 122.4, 119.0 (d, *J* = 14.6 Hz), 111.3 (dd, *J* = 21.7, 3.8 Hz), 103.5 (t, *J* = 25.7 Hz), 97.2, 95.4, 69.6, 64.8 (d, *J* = 3.8 Hz), 59.1, 57.8, 56.6, 49.1, 48.7, 39.0, 37.4, 34.9, 31.5, 28.7, 25.9, 25.7, 22.4, 21.0, 20.0, 15.9, 14.5. HRMS (ESI): calculated for [C_53_H_61_ O_7_N_9_BrF_2_S]^+^: 1084.35606, found 1084.34834 (Appendix A).

#### 2.2.2. (2*S*,4*R*)-1-((*S*)-2-(3-(4-(4-(2-(3-(3-bromo-4-((2,4-difluorobenzyl)oxy)-6-methyl-2-oxopyridin-1(2H)-yl)-4-methylbenzamido)ethyl)-1H-1,2,3-triazol-1-yl)butanamido)propanamido)-3,3-dimethylbutanoyl)-4-hydroxy-*N*-((*S*)-1-(4-(4-methylthiazol-5-yl)phenyl)ethyl)pyrrolidine-2-carboxamide (**NR-11b**)

Compound **NR-11b** was prepared following the general method for CuCAAC, starting from alkyne **1** (40 mg, 0.08 mmol) and (2*S*,4*R*)-1-((*S*)-2-(3-(4-azidobutanamido)propanamido)-3,3-dimethylbutanoyl)-4-hydroxy-*N*-((*S*)-1-(4-(4-methylthiazol-5-yl)phenyl)ethyl)pyrrolidine-2-carboxamide (**2b**, 48 mg, 0.08 mmol). The crude was purified by flash column chromatography, eluting with 7% MeOH/DCM. Compound **NR-11b** was obtained as a white solid (57 mg, 0.05 mmol, 62% yield).

^1^H NMR (400 MHz, CD_3_OD) δ 8.87 (s, 1H), 7.86 (dt, *J* = 8.0, 1.9 Hz, 1H), 7.79 (s, 1H), 7.70–7.59 (m, 2H), 7.50 (d, *J* = 8.0 Hz, 1H), 7.47–7.31 (m, 5H), 7.08–6.99 (m, 2H), 6.65 (d, *J* = 1.1 Hz, 1H), 5.35 (s, 2H), 4.99 (qd, *J* = 7.0, 2.2 Hz, 1H), 4.63–4.52 (m, 2H), 4.39 (q, *J* = 7.1, 6.4 Hz, 3H), 3.93–3.84 (m, 1H), 3.76–3.66 (m, 1H), 3.49–3.32 (m, 4H), 2.76 (t, *J* = 7.4 Hz, 2H), 2.45 (dt, *J* = 6.7, 3.0 Hz, 5H), 2.22–1.88 (m, 12H), 1.48 (dd, *J* = 7.0, 2.4 Hz, 3H), 1.02 (s, 9H) (Appendix A).

^13^C NMR (101 MHz, CD_3_OD) δ 173.1, 172.2, 171.8, 170.8, 167.1, 164.5, 163.4 (d, *J* = 241.8 Hz), 161.0, 160.9 (dd, *J* = 243.2, 12.4 Hz), 151.5, 147.2, 144.3, 139.2, 137.7, 133.8, 131.3, 131.3, 131.2, 131.2, 130.1, 129.1, 129.1, 128.0, 126.6, 126.2, 125.9, 122.3, 119.0 (d, *J* = 14.7 Hz), 111.3 (dd, *J* = 21.6, 3.8 Hz), 103.5 (t, *J* = 25.7 Hz), 97.2, 95.4, 69.6, 64.8 (d, *J* = 3.8 Hz), 59.1, 57.8, 56.5, 49.3, 48.7, 39.0, 37.5, 35.7, 34.9, 34.9, 32.1, 28.7, 25.9, 25.7, 22.4, 21.0, 20.0, 15.9, 14.5 (Appendix A).

HRMS (ESI): calculated for [C_56_H_66_ O_8_N_10_BrF_2_S]^+^: 1155.39318, found 1155.38474.

#### 2.2.3. (2*S*,4*R*)-1-((*S*)-2-(6-(4-(4-(2-(3-(3-bromo-4-((2,4-difluorobenzyl)oxy)-6-methyl-2-oxopyridin-1(2H)-yl)-4-methylbenzamido)ethyl)-1H-1,2,3-triazol-1-yl)butanamido)hexanamido)-3,3-dimethylbutanoyl)-4-hydroxy-*N*-((*S*)-1-(4-(4-methylthiazol-5-yl)phenyl)ethyl)pyrrolidine-2-carboxamide (**NR-11c**)

Compound **NR-11c** was prepared following the general method for CuCAAC, starting from alkyne **1** (35 mg, 0.07 mmol) and (2*S*,4*R*)-1-((*S*)-2-(6-(4-azidobutanamido)hexanamido)-3,3-dimethylbutanoyl)-4-hydroxy-*N*-((*S*)-1-(4-(4-methylthiazol-5-yl)phenyl)ethyl)pyrrolidine-2-carboxamide (**2c**, 45 mg, 0.07 mmol). The crude was purified by flash column chromatography, eluting with 12% MeOH/DCM. Compound **NR-11c** was obtained as a white solid (48 mg, 0.04 mmol, 57% yield).

^1^H NMR (400 MHz, CD_3_OD) δ 8.86 (s, 1H), 7.86 (dt, *J* = 8.0, 1.6 Hz, 1H), 7.77 (s, 1H), 7.71–7.58 (m, 2H), 7.51 (d, *J* = 8.1 Hz, 1H), 7.48–7.34 (m, 4H), 7.04 (t, *J* = 8.6 Hz, 2H), 6.66 (d, *J* = 1.0 Hz, 1H), 5.35 (s, 2H), 4.99 (q, *J* = 7.0 Hz, 1H), 4.65–4.54 (m, 2H), 4.41 (dq, *J* = 4.0, 2.0 Hz, 1H), 4.37 (td, *J* = 5.6, 4.8, 1.8 Hz, 2H), 3.89–3.83 (m, 1H), 3.73 (dd, *J* = 11.0, 4.0 Hz, 1H), 3.40 (td, *J* = 6.9, 2.5 Hz, 2H), 3.13 (t, *J* = 7.0 Hz, 2H), 2.76 (t, *J* = 7.5 Hz, 2H), 2.46 (d, *J* = 6.6 Hz, 3H), 2.26 (p, *J* = 7.2 Hz, 3H), 2.16 (dt, *J* = 5.9, 2.8 Hz, 4H), 2.08 (s, 3H), 2.02–1.90 (m, 6H), 1.60 (dt, *J* = 15.3, 7.6 Hz, 2H), 1.53–1.43 (m, 5H), 1.42–1.25 (m, 2H), 1.01 (d, *J* = 9.7 Hz, 9H) (Appendix A).

^13^C NMR (101 MHz, CD_3_OD) δ 174.4, 172.9, 171.8, 170.9, 167.1, 164.5, 163.5 (d, *J* = 241.6 Hz), 161.0, 160.9 (dd, *J* = 243.5, 12.9 Hz), 151.4, 147.2, 147.0, 144.3, 139.2, 137.7, 133.8, 131.4, 131.3, 131.2, 130.1, 129.1, 128.0, 126.6, 126.2, 122.1, 119.0 (d, *J* = 14.6 Hz), 112.0–109.9, 103.5 (t, *J* = 25.7 Hz), 97.2, 95.4, 69.5, 64.8 (d, *J* = 3.8 Hz), 59.1, 57.6, 56.6, 49.2, 48.7, 38.9, 38.9, 37.4, 35.1, 35.0, 32.1, 28.7, 28.6, 26.2, 26.0, 25.7, 25.2, 22.4, 21.0, 20.0, 15.9, 14.4 (Appendix A).

HRMS (ESI): calculated for [C_59_H_72_O_8_N_10_BrF_2_S]^+^: 1197.44013, found 1197.43677.

#### 2.2.4. (2*S*,4*S*)-1-((*S*)-2-(6-(4-(4-(3-(3-(3-bromo-4-((2,4-difluorobenzyl)oxy)-6-methyl-2-oxopyridin-1(2H)-yl)-4-methylbenzamido)propyl)-1H-1,2,3-triazol-1-yl)butanamido)hexanamido)-3,3-dimethylbutanoyl)-4-hydroxy-*N*-((*S*)-1-(4-(4-methylthiazol-5-yl)phenyl)ethyl)pyrrolidine-2-carboxamide (**NR-11c***)

**NR-11c*** was prepared following the general method for CuCAAC, starting from alkyne **1** (10 mg, 0.019 mmol) and (2*S*,4*S*)-1-((*S*)-2-(6-(4-azidobutanamido)hexanamido)-3,3-dimethylbutanoyl)-4-hydroxy-*N*-((*S*)-1-(4-(4-methylthiazol-5-yl)phenyl)ethyl)pyrrolidine-2-carboxamide (**2c***, 12 mg, 0.019 mmol). The crude was purified by flash column chromatography, eluting with 11% MeOH/DCM. Compound **NR-11c*** was obtained as a white solid (12.5 mg, 0.010 mmol, 55% yield) (Appendix A).

HRMS (ESI): calculated for [C_59_H_72_O_8_N_10_BrF_2_S]^+^: 1197.44013, found 1197.43914.

### 2.3. Cell Culture

MDA-MB-231, T47D, U20S, MG63 and SAOS-2 cell lines were purchased from ATCC. BBL358 cells were generated as described [20]. All cells were maintained in DMEM high glucose supplemented with 10% FBS, 1% L-Glutamine and 1% penicillin-streptomycin at 37°C and 5% CO_2_. Treatments were performed with PROTACs at the indicated concentrations or with 1 μM of PH-797804 (Selleckchem #S2726). To induce p38α pathway activation, cells were stimulated with UV (40 J/m^2^) and collected after 1 h. Cells were treated with 20 μM MG132 (Calbiochem #474790) or 1 μM NAE inhibitor (NEDD8-Activating Enzyme inhibitor, Calbiochem #505477) for 1 h before PROTAC addition. For siRNA transfection, cells were cultured in OPTIMEM, and *MAPK14* siRNA (Ambion, #s3586) was used with Lipofectamine RNAiMAX following the manufacturer’s instructions.

### 2.4. Immunoblotting

Cells or mouse tissues were lysed in RIPA buffer containing 1% NP40, 0.5% sodium deoxycholate, 0.1% SDS, 50mM Tris-HCl, 150 mM NaCl, 5 mM EDTA, 5 mM EGTA, 20 mM sodium fluoride, 1 mM PMSF, 1 mM sodium orthovanadate, 2.5 mM benzamidine, 10 μg/mL pepstatin A, 1 μM mycrocystin, 10 μg/mL leupeptin and 10 μg/mL aprotinin. Tissue samples were homogenized using the Precellys instrument. Lysates were kept on ice for 15 min and then centrifuged at 13.000× *g* for 15 min at 4 °C. Supernatants were recovered for protein quantification and analyzed by immunoblotting. Protein content was determined using the DC protein assay from Bio-Rad (#5000111) following the manufacturer’s instructions. Proteins lysates (30–50 μg) were separated on SDS-PAGE and transferred to nitrocellulose membranes (GE Healthcare life sciences # 10600001). Membranes were blocked in TBST with 10% non-fat milk for 1 h at RT. Incubation with primary antibodies was performed at 4 °C overnight. The following antibodies were used: p38α (Santa cruz #sc-81621; 1:500), p38β (Cell signaling #2339; 1:500), p38γ (Cell signaling #2308; 1:1000), p38δ (Cell signaling #2307; 1:1000), MK2 (Cell signaling #3042; 1:500), phospho-MK2 (Cell signaling #3007; 1:500), Erk1/2 (Cell signaling #9102; 1:1000), Jnk (Santa cruz #sc-7345; 1:2000) and α-tubulin (Sigma #T9026; 1:10000). Secondary antibodies were incubated for 1 h at RT in TBST with 10% non-fat milk. Protein bands were detected using the Odyssey Infrared Imaging System (Li-Cor, Biosciences) and quantified by ImageJ.

### 2.5. Thermal Shift Assays

Samples were prepared in triplicates by mixing purified proteins (1 μg) with the compounds or the equivalent volume of DMSO as a control in Protein Thermal Shift Buffer (1X final concentration, Applied Biosystem Protein Thermal Shift™ Kit #4461146, Carlsbad, CA, USA). Compounds or DMSO were diluted in 20 mM Tris pH 7.5, 100 mM NaCl, 2 mM MgCl_2_, 2 mM DTT. Samples were incubated at RT for 30 min protected from the light. Before reading the plate, 1X SYPRO Orange was added. Fluorescence intensity was measured at 1 °C intervals from 25 °C–95 °C at a rate of 0.05 °C/s using a Viia 7 Real-Time PCR System from Life Technologies.

### 2.6. RNA Extraction and Gene Expression Analysis

For RNA extraction and gene expression analysis, 5 × 10^5^ cells were lysed in 500 μL Trizol and 100 μL chloroform were added to generate two liquid phases after centrifugation at 15.000× *g* for 10 min. The colorless fraction was mixed with ethanol 70% 1:1(*v*/*v*), and the RNA was purified using the PureLink RNA mini kit (Invitrogen Life Technologies #12183018A, Carlsbad, CA, USA). DNAse treatment was performed using on-column DNase treatment (Life Technologies #12185010) following the manufacturer’s instructions. The cDNAs were synthesized using SuperScript IV reverse transcriptase from Invitrogen. The qRT-PCR reactions were performed in triplicates using 12 ng of cDNA and SYBR Green reagent (Life Technologies #4472954) in a final reaction volume of 10 μL containing 200 nM of each primer. RT-PCR experiments were performed with the following protocol on a QuantStudio 6 Flex Instrument (Life Technologies): 50 °C for 2 min, 95 °C for 10 min, 40 cycles of denaturation at 95 °C for 15 s, annealing at 56 °C for 15 s, elongation at 72 °C for 60 s and three final steps of 95 °C for 15 s, 60 °C for 2 min and 95 °C for 15 s. The following primers were used: p38α Fw: ACTCAGATGCCGAAGATGGAAC, p38α Rv: GTGCTCAGGACTCCATCTCT, MK2 Fw: GGATGTCAAGCCTGAGAAT, MK2 Rv: CCAGCACTTCTGGAGCCAC; GAPDH Fw: GGATTTGGTCGTATTGGG, GAPDH Rv: GGAAGATGGTGATGGGATT. The mRNA levels were normalized to the GAPDH expression levels.

### 2.7. Treatment of Mice with PROTACs

Mice (*Mus musculus*) were housed in the specific pathogen-free (SPF) mouse facility of the Barcelona Science Park (PCB, Barcelona). Experiments were performed following the European Union, national and institutional guidelines, and experimental protocols were approved by the Animal Care and Use Committee of the PCB (CEEA-PCB, 21-065). 

C57BL/6 mice (8-10 weeks old males) were administered intraperitoneally or via tail vein injection the PROTAC **NR-11c**, at 15 mg/kg in PBS containing 50% hydroxypropyl β-cyclodextrin. Control mice were treated with the PBS-cyclodextrin vehicle. Tissues were collected at the indicated times, snapped frozen in liquid nitrogen and stored at −80°C. Samples were analyzed by immunoblotting.

For the generation of mammary tumors, 5 × 10^5^ MDA-MB-231 cells were resuspended in PBS and mixed with Matrigel 1:1 (*v*/*v*) in a final volume of 50 μL. Cells were injected into the mammary fat pad of NOD/SCID gamma female mice (8–10 weeks old). When tumors reached 100–150 mm^3^, mice were treated with **NR-11c** via the indicated administration routes using a dose of 15 mg/kg in a volume of 50 μL.

## 3. Results

### 3.1. Design of New VHL-Based PROTACs to Target p38α

We hypothesized that PROTACs based on a VHL ligand could be more soluble than the previously developed ones that were based on a CRBN ligand. Therefore, we designed three compounds based on the VHL ligand and PH-797804, which differ in linker length and were named **NR-11a**, **NR-11b** and **NR-11c** (Figure 1). We observed that **NR-11c** induced potent degradation of p38α in three different breast cancer cell lines to similar levels as the previously reported PROTAC **NR-7h** based on CRBN [18]. In contrast, **NR-11a** and **NR-11b** induced p38α degradation in human MDA-MB-231 and T47D cells, but not in mouse BBL358 cells (Figure 2A). Based on these results, we selected **NR-11c** for further characterization.

To investigate the degradation activity of **NR-11c**, we chose the osteosarcoma cell lines U2OS, MG63 and SAOS-2, in which our previous work showed that p38α degradation was poorly induced by the CRBN-based PROTAC **NR-7h** [18]. Interestingly, **NR-11c** showed a substantially increased ability to decrease p38α protein levels in the three osteosarcoma cell lines compared with **NR-7h** (Figure 2B). These results suggest that PROTACs based on VHL are able to induce p38α degradation in a wider range of cell lines.

### 3.2. Characterization of the PROTAC NR-11c

We estimated the concentration of **NR-11c** required to induce half-maximal degradation of p38α in MDA-MB-231 cells and calculated a DC_50_ value of 11.55 nM (Figure 3A). Consistent with the “hook effect” reported in bifunctional molecules such as PROTACs at higher concentrations due to the binary interactions that compete with the tertiary complex formation [21], we observed some decreased efficiency of p38α degradation at 10 μM (Figure 3A).

Next, we analyzed the treatment time required to induce p38α degradation and found that p38α was fully degraded in just 2–4 h after adding **NR-11c** at 1 μM, and even at 0.1 μM, a substantial reduction in the p38α levels was observed in 4 h (Figure 3B). We also investigated the long-term effect of **NR-11c** and found that p38α degradation was maintained for 72 h after the addition of **NR-11c** to the cells (Figure 3C). Interestingly, we still observed significant degradation of p38α when cells were treated with **NR-11c** for 24 h and then analyzed up to 48 h after removal of the compound from the media (Figure 3C).

To characterize the mechanism of **NR-11c**-induced degradation of p38α, we pre-treated the cells with the proteasome inhibitor MG132 or with the NEDD8-Activating Enzyme (NAE), a compound that prevents activation of cullin-RING ubiquitin ligases. Degradation of p38α was impaired with both compounds, indicating that **NR-11c** indeed targets the p38α protein for degradation by the proteasome pathway in a ubiquitin ligase-dependent manner (Figure 4A). We also confirmed that the inactive isomer **NR-11c***, which is impaired in its ability to engage the E3 ligase, did not induce p38α degradation (Figure 4A). Consistent with these observations, treatment of cells with **NR-11c** did not change the expression levels of the p38α-encoding mRNA *MAPK14* (Figure 4B), supporting that the reduced p38α levels observed upon treatment with this compound were due to protein degradation.

Since p38α has been reported to form a complex with MK2 in mammalian cells, and the downregulation of p38α affects the MK2 protein levels [22,23], we investigated the effect of **NR-11c** treatment on MK2 levels. We treated MDA-MB-231 cells with **NR-11c**, with the inactive compound **NR-11c*** or with a siRNA against p38α, and 24 h later, the MK2 levels were checked both by immunoblotting and by RT-qPCR analysis. As expected, we found that the protein levels of MK2 were decreased with both the p38α-targeting siRNA and **NR-11c** but not with **NR-11c*** (Figure 4C). However, the MK2 mRNA levels did not change with any treatment (Figure 4D). These results indicate that PROTAC **NR-11c** mimics the effects of the lack of p38α induced by other methods such as RNA interference or genetic knockout [22].

We previously showed that CRBN-based PROTACs are able to selectively degrade p38α and p38β but no other p38 MAPKs. To investigate the specificity of the VHL-based PROTACs, we analyzed the levels of the four p38 MAPK family members in MDA-MB-231 and T47D cells after 24 h of PROTAC treatment. Intriguingly, we observed that **NR-11c** specifically degraded p38α but not p38β. As expected, treatment of the same cells with **NR-7h** induced the degradation of both p38α and p38β (Figure 5A). We also confirmed that other MAPKs, namely JNK and ERK1/2, were not affected by treatment with **NR-11c** (Figure 5A).

To investigate if the ability of **NR-11c** to induce the degradation of p38α but not p38β was due to its differential binding to both proteins, we performed thermal shift assays using purified p38α and p38β. Calculation of the melting temperatures, as an indicator of how compound binding affects protein stability, showed that **NR-11c** and **NR-7h** both stabilized p38α to a similar extent as the inhibitor PH-797804, evidenced by a shift to the right in the three denaturation curves compared with the DMSO control. Remarkably, melting temperature values for p38β were also changed similarly by **NR-11c** and **NR-7h** (Figure 5B). Taken together, these results suggest that **NR-11c** has the ability to bind in vitro to both p38α and p38β, but in cells, only the interaction with p38α leads to a productive complex with the E3 ligase that results in its degradation.

To explore potential differences between p38α inhibition by PH-797804 and its degradation induced by **NR-11c**, we stimulated cells with UV to induce activation of the p38α pathway. We found that PH-797804 efficiently impaired the phosphorylation of MK2, an important p38α substrate, without affecting MK2 or p38α expression levels. Likewise, p38α signaling was strongly reduced in cells treated with **NR-11c**, as shown by the reduced MK2 phosphorylation levels. However, in this case, the total levels of both p38α and MK2 were downregulated (Figure 6A). Following these observations, we investigated if the p38α pathway could remain inactive for an extended period of time in cells treated with **NR-11c** compared with PH-797804. Therefore, we treated cells with UV and analyzed the pathway status 24 h after stimulation as well as at different times upon removal of the compounds from the media (washout). We found that in cells treated with **NR-11c**, p38α degradation was maintained for up to 72 h after the washout. However, the inhibitory effect of PH-797804 was substantially reduced upon removal of the compound from the media (Figure 6B). The prolonged time of p38α degradation also correlated with reduced levels of MK2, consistent with the requirement for complex formation between p38α and MK2 for the mutual stabilization of both proteins [22].

### 3.3. Effects of the PROTAC NR-11c In Vivo

Our previously described p38α-targeting PROTACS based on CRBN [18] were rather insoluble in aqueous solvents, which limited the possibility of evaluating their effectivity in vivo. Interestingly, the new PROTAC **NR-11c** based on VHL was completely dissolved in a PBS solution containing cyclodextrin, a carbohydrate widely used for enhancing the aqueous solubility and stability of drugs [24,25]. We administered **NR-11c** to mice either through tail vein injection or intraperitoneally, and 24 h later, we assessed p38α protein levels in different tissues. We found that p38α protein levels were clearly downregulated in the liver from mice treated with **NR-11c** but did not change in lung, kidney or spleen taken from the same mice (Figure 7A). The downregulation of p38α levels in liver was sustained for up to 48 h after PROTAC administration (Figure 7B). These results suggest that **NR-11c** was probably inactivated in the liver, preventing it from reaching other tissues.

In another set of experiments, we evaluated the effect of **NR-11c** after local administration in the context of a tumor. We implanted MDA-MB-231 cells into the fat mammary pads and when the tumors grew to a size of 100 mm^3^, **NR-11c** was injected either into the tumor (intratumoral) or in the surrounding area (peritumoral). Interestingly, we observed p38α downregulation in tumors upon both types of administration (Figure 7C). These observations led us to analyze how **NR-11c** affected p38α expression in tumors at different time points after administration. Mice were administered a single dose of **NR-11c**, and the p38α protein levels in the mammary tumors were analyzed after 24, 48 and 72 h. We found that although **NR-11c** induced a sustained downregulation of p38α at 24 h, the levels were almost recovered after 48 h, indicating a fast p38α protein turnover in the tumors (Figure 7D).

## 4. Discussion

PROTACs are small molecules that transiently interact with a target protein, inducing its ubiquitination by an E3 ligase and subsequent degradation. This catalytic mechanism of action contrasts with the function of other small-molecule drugs that typically bind to the target protein, blocking its activity. Therefore, PROTAC technology has developed rapidly with the promise to find potential solutions to replace inhibitors with low selectivity or to interfere with the activity of noncatalytic proteins [15,17,26].

In this study, we designed PROTACs for targeting the protein kinase p38α based on the ATP-competitor PH-797804 and a ligand to the VHL E3 ubiquitin ligase. This work builds on our previous report showing that PROTACs based on thalidomide for recruiting the CRBN E3 ubiquitin ligase were able to degrade with high potency and efficacy both p38α and p38β family members in mammalian cell lines [18]. However, the CRBN-based PROTACs were poorly soluble in aqueous solutions, which prevented their use in mice. Optimization of the physicochemical properties to turn protein degraders that work in vitro into in vivo drugs represents a big challenge to impulse the use of PROTACs in the clinic. For this reason, we decided to generate a new type of PROTACs to target p38α, which were based on VHL and that we hypothesized would be better solubilized in aqueous media [27]. Our results confirmed that one of these PROTACs, **NR-11c**, can induce p38α degradation both in mammalian cell lines and in mice.

Several PROTACs based on CRBN are being currently tested in clinical trials [28], and it has been proposed that the chemical properties of the CRBN-based PROTACs endow them with higher oral bioavailability and plasma clearance than the VHL-based PROTACs [29]. This suggests that CRBN-based PROTACs could be more suitable for therapeutic applications. However, it has been recently shown that VHL-based PROTACs are also suitable for oral administration [30,31] and can be combined with various drug delivery methods [32]. A VHL-based PROTACs has also been recently reported to reach the brain [33]. Interestingly, a study comparing VHL and CRBN dependencies in diverse tumor types concluded that VHL-based PROTACs can induce degradation in a broader number of cell lines, as CRBN was frequently found inactivated or expressed at low levels in lung and colon cancer lines [34]. Our results confirm that PROTACs based on VHL or CRBN have different activities depending on the cell line, with VHL-based PROTACs usually having wider activity. In addition, we show that local administration of PROTACs successfully induces p38α degradation in tumors, and this type of administration is being evaluated in clinical trials with immunotherapeutic drugs [35].

An important topic in the field of PROTACs is the specificity given by the linker, warhead or E3 ubiquitin ligase ligand. We previously reported that longer linkers improve the performance of CRBN-based PROTACs [18]. Another study showed that PROTACs with distinct linker attachment sites and lengths can trigger the selective degradation of p38α versus p38δ [36]. Interestingly, our results show that the PROTAC specificity for p38β can be modified by changing the E3 ubiquitin ligase. We found that **NR-11c** can bind to both p38α and p38β in vitro, but it can only induce the degradation of p38α in cells. This suggests the existence of factors that affect either the interaction of the PROTAC with p38α and p38β in cells or the affinities for the E3 ligase complex, resulting in the selective degradation of p38α but not p38β. Previous publications have highlighted the importance of the ternary complex formation to stabilize the interactions between the E3 ubiquitin ligase and the target protein [37]. It has been also reported that PROTACs exhibit a higher selectivity in comparison with the warhead used for their design, at least in part due to the protein–protein interactions that define the stability of the complexes [38,39]. PROTACs based either on CRBN or on VHL have been reported to differentially induce the degradation of BCR/Abl or c-Abl proteins [40], indicating that the E3 ligase recruited is a crucial component to determine the PROTAC specificity.

The activity of PROTACs has been widely studied in cell lines, but there is limited information on their pharmacokinetic properties, which is critical to understand the in vivo efficiency of these molecules. Using cryopreserved human hepatocytes, it has been reported that the linker’s chemical nature and length are major determinants of PROTAC metabolic instability [41]. For instance, short linkers provide more stability compared with longer ones [41], but long linkers tend to improve the activity and are commonly used in PROTAC design [18]. We propose that the absence of effect of **NR-11c** in tissues other than liver upon systemic administration to mice could be due to its inactivation in the liver. However, the in vivo activity of a PROTAC may be influenced by several factors beyond its pharmacokinetic properties, such as the ability of the PROTAC to reach a tissue, the levels of proteasome and E3 activity in particular cells or the re-synthesis rate and stability of the target protein [21,42]. Since these factors may differentially affect various PROTACs, further work will be needed to understand how PROTAC-induced p38α degradation is controlled in vivo.

Interestingly, comparison of the PROTAC with the chemical inhibitor in which the PROTAC was based indicates that the PROTAC tends to have a more sustained inhibitory effect, affecting not only the degradation of the p38α protein but also of its substrate MK2. We have reported that the p38α-MK2 complex is crucial in determining cell survival in stress situations [22], and the use of PROTACs allowed us to confirm the dependency of each protein on the stability of the other one. Moreover, cells treated with PROTACs would be expected to have a stronger phenotype compared with the chemical inhibitor-treated cell in situations where the cell function depends on the p38α-MK2 signaling axis.

## 5. Conclusions

In summary, we describe a series of new PROTACs that specifically induce the degradation of the p38α protein at nanomolar concentrations and that maintain low levels of this protein after the compound is removed from the cell culture. We also show that these PROTACs are soluble in aqueous solution and can be used in mice. Further studies will be needed to evaluate its potential therapeutic applications, including its pharmacodynamic and pharmacokinetic properties and the best delivery strategies in the clinical context.

## Figures and Tables

**Figure 1 cancers-15-00611-f001:**
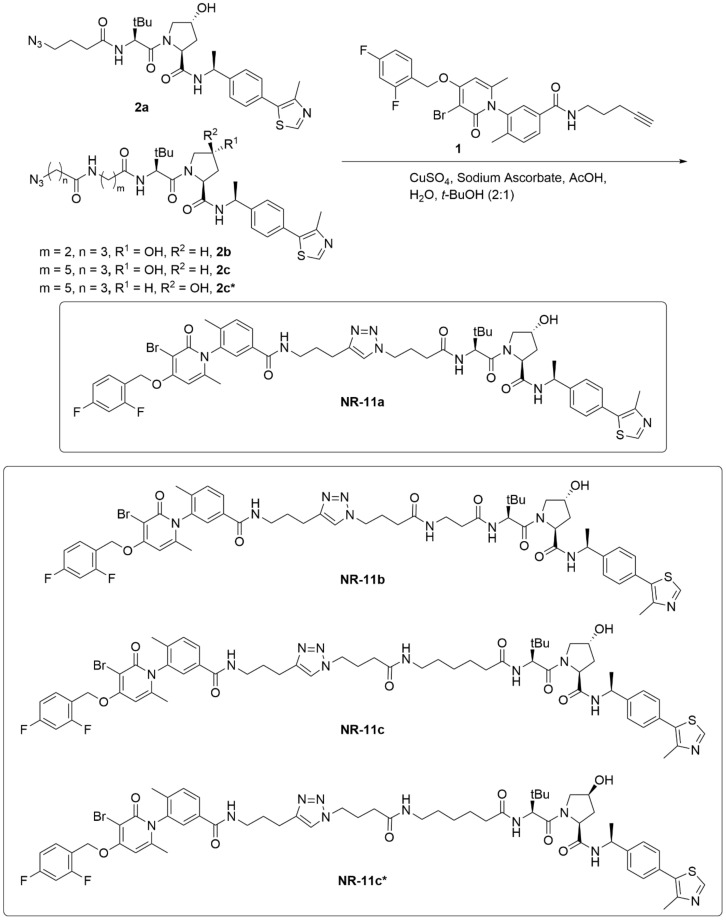
Synthesis of PROTACS **NR-11a–c**.

**Figure 2 cancers-15-00611-f002:**
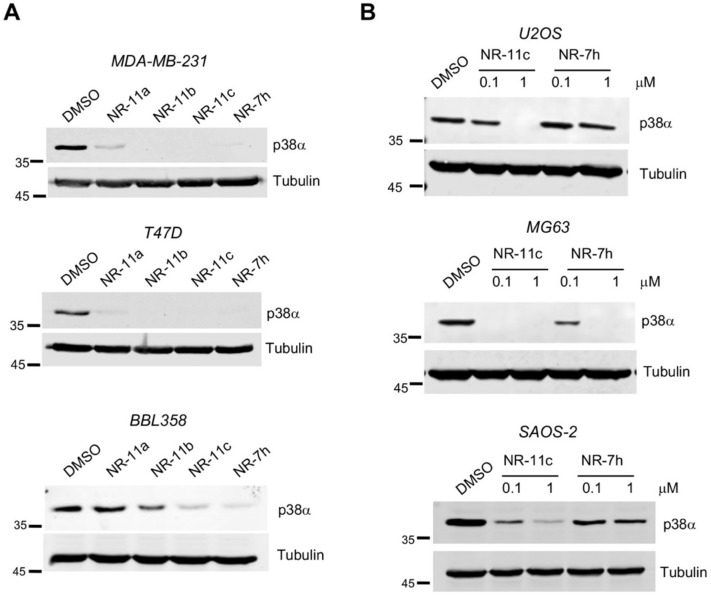
VHL-based PROTACs that target p38α for degradation. (**A**) The indicated breast cancer cell lines were incubated with the indicated compounds at 1 μM, and 24 h later, p38α was analyzed in cells lysates by immunoblotting. (**B**) The indicated osteosarcoma cell lines were incubated with **NR-11c** and **NR-7h** at 1 or 0.1 μM for 24 h and p38α was analyzed in cells lysates by immunoblotting. Immunoblots were performed with a minimum of *n* = 2 biological replicates with similar results obtained each time. The uncropped blots are shown in Appendix A.

**Figure 3 cancers-15-00611-f003:**
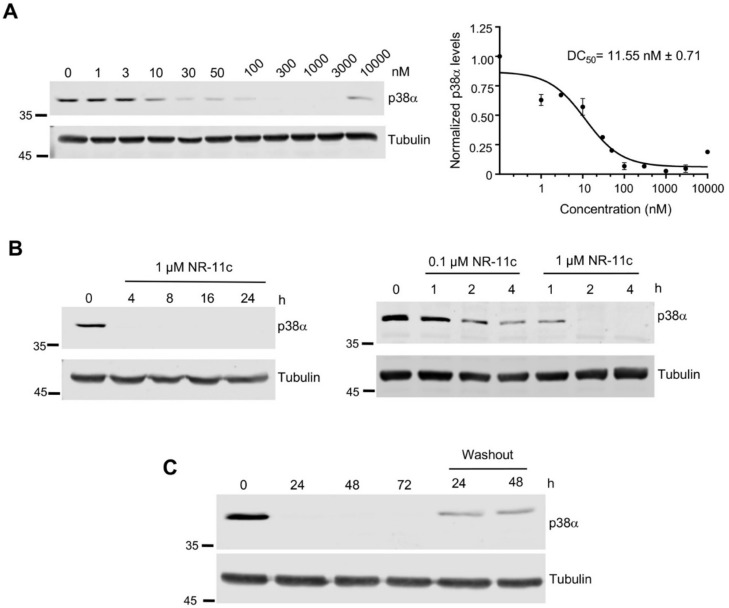
**NR-11c** efficiently induces p38α degradation. (**A**). MDA-MB-231 cells were treated with the indicated concentrations of **NR-11c** for 24 h, and then cell lysates were analyzed by immunoblotting. DC_50_ values for p38α degradation were determined based on the quantification of the p38α band. (**B**). MDA-MB-231 cells were treated with **NR-11c** at 1 or 0.1 μM for the indicated times, and p38α levels were evaluated by immunoblotting. (**C**). MDA-MB-231 cells were cultured with **NR-11c** at 1 μM for the indicated times, or after 24 h, the compound was removed, and cells were maintained for another 24 or 48 h without the compound (washout). Cells lysates were analyzed by immunoblotting. Immunoblots were performed with a minimum of *n* = 2 biological replicates with similar results obtained each time. The uncropped blots are shown in Appendix A.

**Figure 4 cancers-15-00611-f004:**
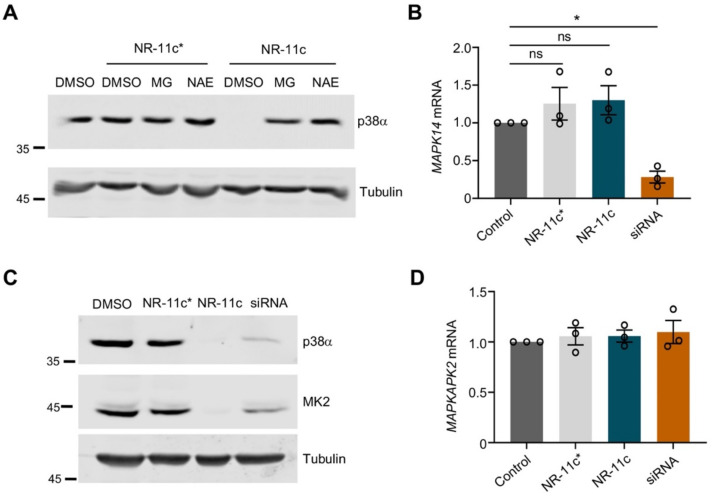
**NR-11c** induces degradation of p38α in a proteasome-dependent manner. (**A**) MDA-MB-231 cells were pre-treated with the proteasome inhibitor MG132 (MG) at 20 μM or the NAE inhibitor at 1 μM for 1 h prior to incubation with either **NR-11c** or the inactive compound **NR-11c*** at 1 μM for 8 h. Cells were collected, and p38α levels were analyzed by immunoblotting. (**B**) MDA-MB-231 cells were transfected with siRNA against the p38α-encoding mRNA *MAPK14* and were analyzed 48 h later or treated for 24 h with the indicated compounds at 1 μM or with DMSO, and then RNA was extracted and was subjected to RT-qPCR using primers specific for *MAPK14*. GAPDH was used for normalization. (**C**) MDA-MB-231 cells were transfected with siRNA against p38α or treated with the indicated compounds and analyzed after 48 or 24 h, respectively. Cell lysates were analyzed by immunoblotting using the indicated antibodies. (**D**) MDA-MB-231 cells were treated as in (**B**), and the purified RNA was subjected to RT-qPCR using primers specific for *MAPKAPK2* (MK2) or for GAPDH for normalization. Immunoblotting experiments were performed with a minimum of *n* = 2 biological replicates and similar results were obtained each time. Statistical analyses were performed using ANOVA and GraphPad Prism Software 9.4.1. * *p* < 0.05; ns, not significant. The uncropped blots are shown in Appendix A.

**Figure 5 cancers-15-00611-f005:**
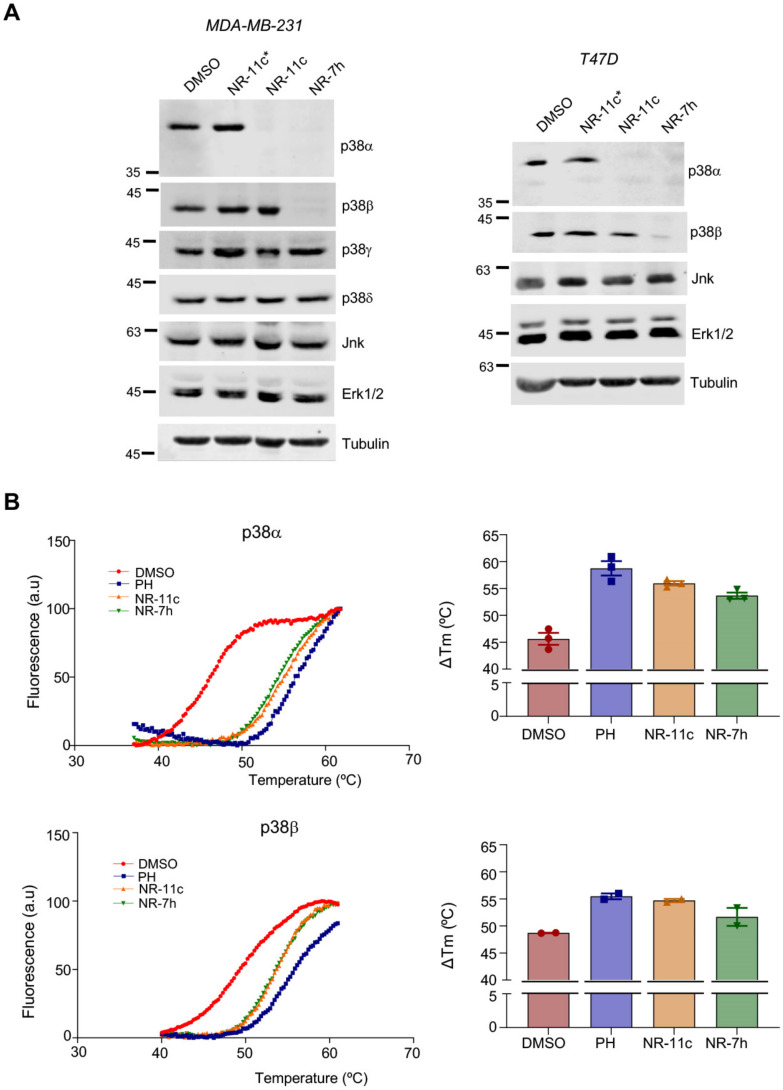
**NR-11c** specifically induces p38α degradation. (**A**) MDA-MB-231 cell and T47D cells were treated with DMSO or the indicated PROTACS at 1 μM for 24 h and cell lysates were then analyzed by immunoblotting using the indicated antibodies. (**B**) Purified p38α and p38β proteins were incubated in the presence of PH-797804 (PH), **NR-11c** or **NR-7h** at 10 μM, and thermal shift assays were performed. Left panels show temperature-dependent fluorescent curves obtained with p38α or p38β. Histograms (right) represent the melting temperature values. Experiments were performed with a minimum of *n* = 2 biological replicates with similar results obtained each time. The uncropped blots are shown in Appendix A.

**Figure 6 cancers-15-00611-f006:**
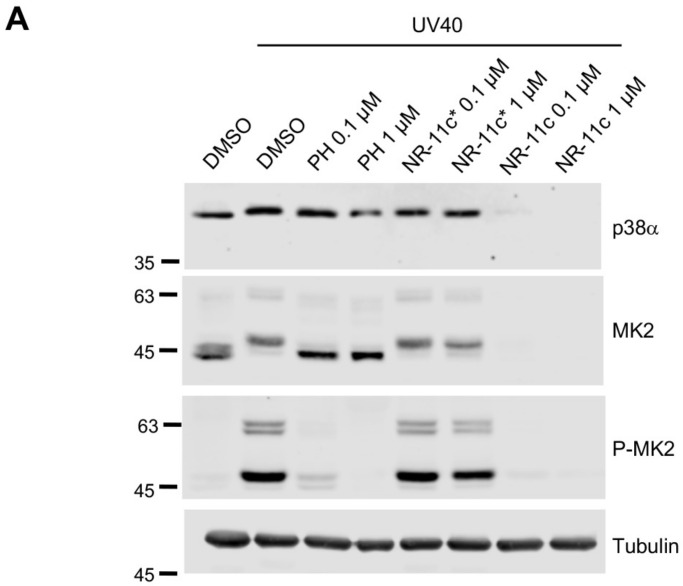
Effect of **NR-11c** on stress-induced p38α activation. (**A**) MDA-MB-231 cells were pre-treated with the inhibitor PH-797804 (PH) or the indicated PROTACs at 1 and 0.1 μM, or with DMSO, for 24 h prior to stimulation with UV-C (40 J/m^2^). After 1 h, cell lysates were analyzed by immunoblotting using the indicated antibodies. (**B**) MDA-MB-231 cells were pre-treated with DMSO, PH or **NR-11c** at 1 μM and 24 h later were stimulated with UV-C. For the washout samples, cells were washed to remove the compounds after 24 h of treatment and then incubated in compound-free media for 24, 48 or 72 h before stimulation with UV-C. In all cases, cell lysates were prepared 1 h after UV stimulation and analyzed by immunoblotting using the indicated antibodies. Experiments were performed with a minimum of *n* = 2 biological replicates with similar results obtained each time. The uncropped blots are shown in Appendix A.

**Figure 7 cancers-15-00611-f007:**
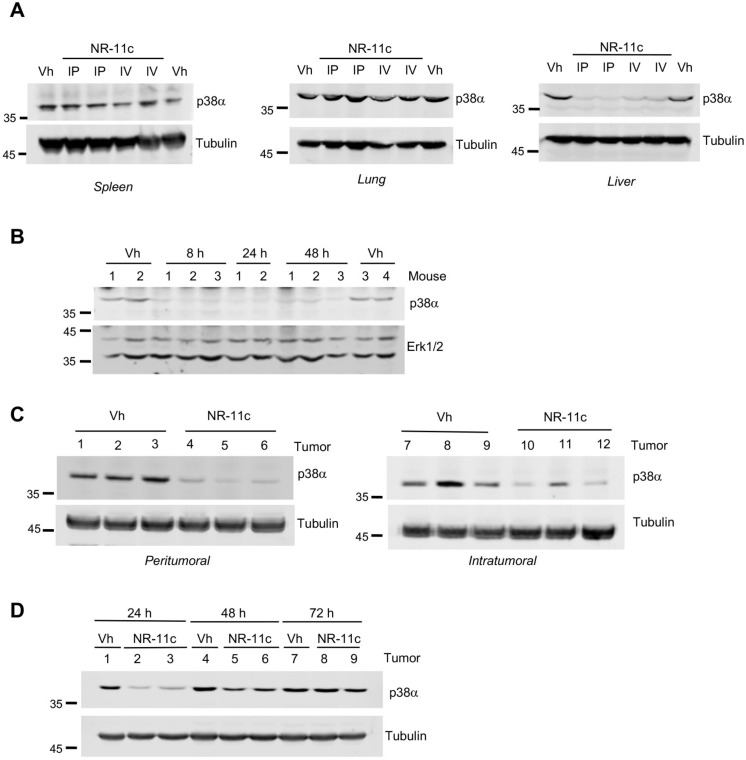
Effect of **NR-11c** in mice. (**A**) C57BL/6J mice were administered intraperitoneal (IP) or intravenously (IV) with **NR-11c** at 15 mg/kg dissolved in PBS with 50% cyclodextrin or with vehicle (Vh). Mice treated via IP received two doses in 24 h, while mice treated via IV received only one dose. After 24 h, the indicated tissues were obtained, and p38α protein levels were analyzed by immunoblotting. Tubulin was used as a loading control. (**B**) C57BL/6J mice were administered IP with **NR-11c** at 15 mg/kg or with Vh, and livers were collected at the indicated times for immunoblotting analysis. Erk1/2 was used as a loading control. (**C**) MDA-MB-231 cells implanted in the mammary fat pad of SCID/NOD mice and allowed to grow up to 100–150 mm^3^ (4–5 weeks). **NR-11c** or Vh were administered via the indicated routes. After 24 h, mammary tumors were collected and p38α protein levels were analyzed by immunoblotting. (**D**) Mice with tumors formed by MDA-MB-231 cells as in (**D**), were injected peritumorally with **NR-11c** or Vh as control, and at the indicated times, tumors were collected and p38α protein levels were analyzed by immunoblotting. All treatments were done with at least 2 mice with similar results obtained in each case. The uncropped blots are shown in Appendix A.

## Data Availability

The data presented in this study are available in this article and Appendix A.

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
