# Peer review of "Synthesis and Biological Activity of a VHL-Based PROTAC Specific for p38α"

_cancers, 2023, doi:10.3390/cancers15030611_

Round 1

Reviewer 1 Report

Overall, this is a well-written manuscript reporting a new PROTAC compound. Particularly, the authors showed the specificity of the compound to p38a in cell lines, although it is unclear whether such selectivity retains in vivo. It would be helpful if the authors can provide some evidence. Apart from this, there are just few minor things:

Line 74 should refers to ref 18?

Please check the degree symbol as many of them have a line underneath the dot.

No need to show the structure of NR-11a/b/c/ in Fig 2A as there are in Fig 1 already.

Please show the structure of NR-11c (for example in Fig 1).

Please align the x-axis of Fig 4D (similar to that of Fig 4B).

Available evidence does not unambiguously support the claim that NR-11c is inactivated in the lever. It is possible that NR-11c are preferentially taken up by liver cells.

Reviewer 2 Report

This study introduced a VHL-based PROTAC specific for p38α and demonstrated it induces degradation of p38α in cultured cells and in vivo. Overall, the experimental design and results are clear and interesting to readers. The manuscript should be accepted for publication in the journal if a major revision is well addressed. Below are some specific comments:

1.       In Figure 4B, the authors didn’t clarify which is the target of siRNA. If it is MAPK14 siRNA, we should observe the mRNA decrease; if not, I would doubt the efficacy of the siRNA or the specificity of the qPCR primers.

2.       In Figure 4D, there is a slight decrease in the MAPKAPK2 mRNA level, but the authors claimed “the MK2 mRNA levels did not change with any treatment (Figure 4D)” in the main text. Is it possible to repeat this data at least 3 times to confirm the conclusion?

3.       NR-11c induced p38α degradation only in the mice liver, and authors deduced that the PROTAC becoming inactivated in that organ. If it is inactivated in the liver, it is hard to imagine the degradation effects can last as long as 48 hours (Figure 7B). Many factors can affect the biodistribution of NR-11c, it’s better to rephrase these possible reasons.

4.       In the abstract, the authors claimed “We report a series of small molecule proteolysis-targeting chimeras (PROTACs) that selectively target the protein kinase p38α for degradation.” But there is not enough evidence displaying the selectivity of NR-11c, is it possible to include a culture cell-based proteomics measurement to show the selectivity of NR-11c at the proteome-wide level?

Reviewer 3 Report

Cubillos-Rojas et al. developed VHL-based PROTAC against p38α protein. The authors studied the degradation of protein kinase p38α after the treatment of synthesized PROTACs. The selectivity profile of these PROTACs was also examined. The authors also confirmed the aqueous solubility of synthesized compounds, showing good bioavailability after administration in mice.

I have some major concerns which need to be addressed before accepting this manuscript for publication in Cancers

1.     The authors are advised to provide NMR spectra of target compounds in the supplementary material.

2.      The authors are also suggested to produce HPLC purity data of the target compounds

3.     Invitro anti-cancer activity of these compounds should also be investigated, using cytotoxicity study in cancer cells

4.     Authors should give anti-tumor activity results of these PROTACs (tumor weight and tumor volume)

5.     Authors have tested in vivo efficacy of these compounds using a single dose for 24, 48, and 72 h. Single-dose treatment is not sufficient to authenticate the effectiveness of drugs. A dose-response study should be conducted using various doses of compounds. Duration of compound treatment to mice should also be increased (at least up to 7 days) to study the time-dependent efficacy of compounds.

6.     There is no toxicity study has been performed. A sub-acute/acute toxicity study of these new molecules should be investigated in mice/rats. 

Round 2

Reviewer 2 Report

my comments have been well addressed, the manuscript should be fine for a publication

Reviewer 3 Report

The authors have revised the ms, incorporating reviewer suggestions. This ms may be accepted for publication.